# Comparing Performance of Spectral Image Analysis Approaches for Detection of Cellular Signals in Time-Lapse Hyperspectral Imaging Fluorescence Excitation-Scanning Microscopy

**DOI:** 10.3390/bioengineering10060642

**Published:** 2023-05-25

**Authors:** Marina Parker, Naga S. Annamdevula, Donald Pleshinger, Zara Ijaz, Josephine Jalkh, Raymond Penn, Deepak Deshpande, Thomas C. Rich, Silas J. Leavesley

**Affiliations:** 1Department of Chemical and Biomolecular Engineering, University of South Alabama, 150 Student Services Dr., Mobile, AL 36688, USA; 2Department of Systems Engineering, University of South Alabama, 150 Student Services Dr., Mobile, AL 36688, USA; 3Department of Pharmacology, University of South Alabama, 5851 USA Drive N., Mobile, AL 36688, USA; annamdevula@southalabama.edu (N.S.A.);; 4Center for Lung Biology, University of South Alabama, 5851 USA Drive N., Mobile, AL 36688, USA; 5College of Medicine, University of South Alabama, 5851 USA Drive N., Mobile, AL 36688, USA; 6College of Medicine, Thomas Jefferson University, Philadelphia, PA 19107, USA

**Keywords:** hyperspectral imaging, fluorescence microscopy, theoretical sensitivity, cellular autofluorescence

## Abstract

Hyperspectral imaging (HSI) technology has been applied in a range of fields for target detection and mixture analysis. While HSI was originally developed for remote sensing applications, modern uses include agriculture, historical document authentication, and medicine. HSI has also shown great utility in fluorescence microscopy. However, traditional fluorescence microscopy HSI systems have suffered from limited signal strength due to the need to filter or disperse the emitted light across many spectral bands. We have previously demonstrated that sampling the fluorescence excitation spectrum may provide an alternative approach with improved signal strength. Here, we report on the use of excitation-scanning HSI for dynamic cell signaling studies—in this case, the study of the second messenger Ca^2+^. Time-lapse excitation-scanning HSI data of Ca^2+^ signals in human airway smooth muscle cells (HASMCs) were acquired and analyzed using four spectral analysis algorithms: linear unmixing (LU), spectral angle mapper (SAM), constrained energy minimization (CEM), and matched filter (MF), and the performances were compared. Results indicate that LU and MF provided similar linear responses to increasing Ca^2+^ and could both be effectively used for excitation-scanning HSI. A theoretical sensitivity framework was used to enable the filtering of analyzed images to reject pixels with signals below a minimum detectable limit. The results indicated that subtle kinetic features might be revealed through pixel filtering. Overall, the results suggest that excitation-scanning HSI can be employed for kinetic measurements of cell signals or other dynamic cellular events and that the selection of an appropriate analysis algorithm and pixel filtering may aid in the extraction of quantitative signal traces. These approaches may be especially helpful for cases where the signal of interest is masked by strong cellular autofluorescence or other competing signals.

## 1. Introduction

Hyperspectral imaging (HSI) technologies were originally developed by NASA and the United States Department of Defense for Earth resource monitoring and military applications [1,2]. More recently, HSI technologies have been applied to a wide range of fields, including document preservation [3,4], agriculture monitoring [5], and medicine [6]. Within the field of biological cellular imaging, HSI technologies have displayed great potential for use with fluorescence microscopy [2,7,8]. In specific, HSI has enabled the detection of many fluorescent labels simultaneously [9,10,11]; the separation of fluorescence signals from cellular and tissue autofluorescence [11,12,13]; and the quantitation of signals, such as for measurement of ratiometric Förster resonance energy transfer (FRET) reporters [14,15,16].

A range of technologies and approaches have been developed in adapting HSI to clinical and biomedical research applications. If the imaging application presents a strong optical signal, high contrast, and does not require high-speed or time-sensitive imaging, a straight-forward approach may be to apply a pre-existing or semi-customized HSI camera or spectral filtering module, along with customized analysis algorithms. This approach is well-suited for transmitted or reflected light HSI, where broad-band illumination may be used to enable a wide spectral scanning range. For example, imaging of fixed and stained histology slides, which present a bright transmitted light signal with negligible photobleaching, can be imaged using a spectral detector consisting of a camera and tunable filter, such as a liquid-crystal tunable filter (LCTF) [17,18,19], acousto-optic tunable filter (AOTF) [20,21], or thin-film tunable filter (TFTF) [22]. Li and colleagues have implemented this approach on an upright microscope platform with illumination provided by a standard microscope lamp (presumed to be a halogen bulb) and spectral detection provided by an AOTF module coupled with a charge-coupled device (CCD) camera [23]. This system provided an HSI range of 550–1000 nm and allowed segmentation and spectral analysis of red blood cell (RBC) smears. The same system was later utilized by the group in a manuscript by Wang et al. for HSI-based studies of melanoma pathology slides, where a convolutional neural network (CNN) analysis was developed to allow segmentation of pathology images as well as automated detection of melanomas, with an overall accuracy of 92% [24]. Finally, HSI datasets of pathology slides of bile duct tissues were acquired using this same system, analyzed with deep neural network (DNN) approaches for detection of cholangiocarcinoma, and found to achieve a patch-level classification accuracy of 94% using the ResNet50 algorithm [25].

In other scenarios, a sample may present a sensitive fluorescence signal that is prone to photobleaching or a live cell, tissue, or in vivo sample may require dynamic imaging to measure a range of cell signaling or physiologic processes. In these cases, there are inherent trade-offs or limitations when implementing HSI for advanced microscopy applications [26,27]. These limitations primarily occur due to the need to divide the photon budget—the fluorescence emission signal—among several dimensions: spatial (X, Y, and in some instances, Z), temporal for time-dependent studies, and spectral (λ). In many cases, the selection of a fluorescence microscope platform and corresponding spectral filtering is determined so as to optimize the imaging performance and shift the weight of the trade-off for a specific application. For example, if high-speed imaging is required for kinetic cellular signaling studies, a widefield or spinning disk confocal microscope may be used where only one or several axial images are acquired, and only one or several wavelength bands are sampled [28,29]. By sampling only one or a minimal number of axial slices and wavelength bands, the temporal sampling may be increased at the expense of axial or spectral sampling. However, there are many biological imaging scenarios where multiple requirements must be considered, such as performing cell signaling studies in cell lines or tissues that are inherently autofluorescent, where both temporal sampling and spectral sampling are needed [30,31]. In these cases, the selection of an appropriate imaging platform and acquisition settings must be performed carefully in order to achieve optimal microscope settings for the study. Post-acquisition image processing may also be applied in an effort to mitigate imaging system trade-offs [32]. Often there is no perfect solution.

We, and others, have previously demonstrated that an alternative approach for hyperspectral imaging—where spectral filtering is implemented in the excitation optics rather than the emission optics, as shown in the optical light path in Figure 1—may provide an increased signal for some studies [33,34,35]. In traditional emission-scanning HSI fluorescence microscope systems (Figure 1A), illumination is typically provided by a broad-band source and a band-pass excitation filter, and the fluorescence emission spectrum is scanned using a tunable filter and camera or a dispersion-based spectral detector based on a grating [36] or prism [37]. A typical emission-scanning experiment would utilize spectral settings so as to excite one or several fluorescence labels near the corresponding excitation spectral peak wavelength(s) and acquire images of fluorescence emission sampled across many spectral bands using a narrow wavelength band for each (Figure 1B). By contrast, in excitation-scanning HSI fluorescence microscope systems (Figure 1C), narrow-band illumination is provided that can be tuned over many different excitation wavelengths while the fluorescence emission is detected using a broad-band or long-pass emission filter and camera. A typical excitation-scanning experiment would utilize spectral settings so as to scan the fluorescence excitation over the range of excitation spectral peak wavelengths while sequentially acquiring an image of fluorescence emission for each excitation wavelength scanned (Figure 1D). Hence, the excitation-scanning HSI approach utilizes the wavelength-dependent properties of the excitation spectrum of each fluorophore in order to perform spectral separation. Because the fluorescence emission is minimally processed in the spectral dimension, the signal reaching the detector is stronger, enabling increased imaging speeds and/or decreased photobleaching.

The overall goal of this study is to compare the performance of common spectral analysis algorithms for use in kinetic cell signaling experiments where excitation-scanning HSI is implemented to enable the separation of the fluorescent label of interest from cellular autofluorescence. Specifically, the goal is to accurately identify a fluorescent label for intracellular Ca^2+^ and to separate this signal from autofluorescence in time-lapse excitation-scanning HSI image data. Recent studies highlight the significance of signal localization in differentiating contractile from non-contractile agonists [38,39,40,41,42]. However, non-uniform cellular autofluorescence can complicate these studies, and HSI approaches offer the potential to separate autofluorescence from Ca^2+^ and other labels. Here, we compare four spectral analysis algorithms: linear unmixing (LU) [43,44], constrained energy minimization (CEM) [45], matched filtering (MF) [46,47], and spectral angle mapper (SAM) [48]. In addition, the effects of using signal intensity thresholding and pixel filtering are evaluated, which enables the analysis of selected pixels above a defined minimum detectable limit. The results indicate that utilizing an optimal spectral analysis algorithm, as well as pixel filtering, can provide an improved ability to quantify weak transient signals, or signals from small regions of the image or localized subcellular regions.

## 2. Materials and Methods

### 2.1. Cell Culture

Human airway smooth muscle cells (HASMCs) were selected as a model system because of the critical role that localized Ca^2+^ signals play in maintaining a contractile state. Recent studies highlight the significance of signal localization in segregating contractile from non-contractile agonists [38,39,40,41,42]. HASMCs were isolated and cultured as described previously [49]. Briefly, HASMCs were grown in 100 mm culture dishes and were maintained in Dulbecco’s Modified Eagles Medium (DMEM, GIBCO) supplemented with 5% fetal bovine serum (Gemini), basic fibroblast growth factor (SIGMA), epidermal growth factor (Invitrogen), 100 U/mL penicillin, and 100 μg/mL streptomycin (Gibco) at pH 7.0. HASMCs were seeded onto 20 mm laminin-coated round glass coverslips and were incubated 37 °C and 5% CO_2_ for 48 h (or until the cells were grown to 70–80% confluency).

Samples with a single label of either NucBlue (nuclear label, ThermoFisher Scientific, Inc., Waltham, MA, USA), Cal 520-AM (Ca^+2^ indicator, AAT Bioquest, Inc., Pleasanton, CA, USA), or unlabeled cells for cellular autofluorescence validation were prepared according to manufacturer labeling specifications. Briefly, single-label NucBlue samples were prepared using 2 drops of NucBlue followed by incubation at 37 °C for 20 min. Single-label Cal 520-AM samples were prepared using 5 µM Cal 520-AM followed by incubation at 37 °C for 30 min. Mixed-label samples were prepared for time-lapse measurements using both Cal 520-AM and NucBlue labeling, using identical labeling concentrations and incubation times as the single-label control samples. After labeling, coverslips were transferred to an attoflour holder (ThermoFisher) for imaging and covered with an extracellular buffer containing (mM): 145 NaCl, 4 KCl, 10 HEPES, 10 D-glucose, 1 MgCl_2_, 1 CaCl_2_, pH 7.4.

### 2.2. Hyperspectral Imaging Acquisition

Excitation-scanning hyperspectral images were acquired utilizing a custom TE-2000 inverted widefield fluorescence microscope (Nikon Instruments, Melville, NY, USA) equipped with 20X objective (Plan Apo 20X, N/A 0.75, Nikon Instruments), Titan 300 Xenon arc lamp (Sunoptics Surgical, Jacksonville, FL, USA), and Prime 95B sCMOS camera (Teledyne Photometrics, Tucson, AZ, USA) as previously described [22,33]. The image acquisition parameters are described in Table 1. A set of thin-film tunable filters (VersaChrome, Semrock, IDEX Health & Science LLC, Rochester, NY, USA) were mounted in a custom tiltable filter wheel (VF-5, Sutter Instrument Co., Novato, CA, USA) that allowed switching between individual TFTFs and tuning of TFTFs by adjusting the angle of the filter wheel. Hyperspectral image data were acquired by sequentially scanning excitation wavelengths from 360 to 480 nm in 5 nm increments while detecting the fluorescence emission at each excitation wavelength using a long-pass dichroic beamsplitter (FF495-Di03, Semrock) and corresponding long-pass emission filter (FF01-496/LP, Semrock). For single-label control samples, only a single hyperspectral image of each sample was acquired. For dynamic Ca^2+^ studies, time-lapse hyperspectral images were acquired by acquiring an excitation-scanning hyperspectral image every 30 s for 15 min. After 5 min of baseline acquisition, cells were treated with either 50 µM carbachol or vehicle control. Spectral image stacks were exported as individual tiff images using NIS Elements software (Nikon Instruments) for subsequent image analysis.

### 2.3. Hyperspectral Image Data Preprocessing

To account for background and wavelength-dependent attenuation, spectral image data were corrected to a NIST-traceable spectral response, as described previously [33]. In brief, a microscope slide-configured laser power meter (ArgoPower, Argolight SA, Pessac, France) was used to measure the illumination power at each excitation wavelength, and the resulting illumination power vs. wavelength data were used as the excitation spectral power and saved as an array, $\vec{\Phi }$. The inverse of the illumination power vs. wavelength data series was normalized to a peak value of unity and used as a spectral correction coefficient.
(1)$$\vec{cc}=\frac{{\vec{\Phi }}^{-1}}{max{\vec{\Phi }}^{-1}}$$

Spectral images were corrected by background subtraction and multiplying by the spectral correction coefficient.

### 2.4. Reference Spectra and Spectral Library

Reference spectra were identified from single-label samples for each fluorescent species: Cal 520, NucBlue, and cellular autofluorescence (unlabeled cells). Spectral images were acquired for each single-label sample, and images were corrected to a flat spectral response, as described above. Regions of interest were then selected corresponding to areas of intense, but not oversaturated, signals using the freehand selection region tool in ImageJ software [50]. The pixel-averaged spectrum from each region was extracted using the Image → Stacks → Plot *Z*-axis Profile command in ImageJ. This served as the reference spectrum for each label. Reference spectra were compiled in Excel (Microsoft Corporation, Redmond, WA, USA) and subsequently transferred to MATLAB (The MathWorks, Inc., Natick, MA, USA) as a spectral library for use with the analysis algorithms described below.

### 2.5. Spectral Analysis Algorithms

#### 2.5.1. Linear Unmixing

Linear unmixing (LU) is a standard approach used by the fluorescence microscopy spectral imaging community for estimating endmember abundances [43,51]. LU enables estimation of the abundance of each endmember present in each pixel, typically using least-squares regression—for fluorescence microscopy, this corresponds to an estimation of the relative signal from each fluorescent label. To utilize linear unmixing, control samples were prepared and imaged to build a spectral library a priori. Linear unmixing was implemented using a modified version of the “lsqnonneg” function in MATLAB, which utilizes a non-negatively constrained least-squares unmixing approach [43,44] (Equation (2)).
(2)$$\vec{x}=\sum_{i=1}^{m}a_{i}\vec{r_{i}}+\vec{w}$$
where $\vec{x}$ is the detected pixel spectrum vector, $a_{i}$ is the endmember, $\vec{r_{i}}$ is the respective endmember spectrum and $\vec{w}$ is the additive observation noise vector.

#### 2.5.2. Spectral Angle Mapper

Spectral angle mapper (SAM) is an algorithm that measures the spectral similarities by finding the angle between the spectral signatures of two pixel vectors and has been widely used in remote sensing due to its advantage of only requiring the target endmember spectrum to be known [48,51]. The angle between pixels and the reference spectrum is then used to classify the image, usually by applying a global threshold. SAM was implemented using a custom MATLAB script to calculate the arccosine of the dot product of the spectra between the measured pixel spectrum and the *i*th endmember spectrum (Equation (3))
(3)$$\theta_{i}=\cos^{-1}\left(\frac{\vec{r}\cdot \vec{x}}{∥\vec{r_{i}}∥\cdot ∥\vec{x}∥}\right)$$
where $\vec{r}$ is the test spectrum and $\vec{x}$ is the reference spectrum.

#### 2.5.3. Constrained Energy Minimization

Constrained energy minimization (CEM) is an algorithm that applies a mathematical filter to amplify the desired spectral signature while minimizing the output energy (e.g., background noise) resulting from target sources other than the desired target [45,52]. CEM utilizes an optical linear operator, $L_{CEM}$ (Equation (4)), to minimize the intensity of undesired signatures while intensifying desired signatures. When using CEM, only the target endmember signature needs to be known. However, CEM presents limitations as it is a single-target detection method and therefore does not allow the preprocessing of known undesired targets for detection enhancement.
(4)$$L_{CEM}=R_{corr}^{-1}\vec{r_{i}}{\left({\vec{r_{i}}}^{T}R_{corr}^{-1}\vec{r_{i}}\right)}^{-1}$$
where the inverse of the sample correlation matrix of the observation pixel vectors ($R_{corr}^{-1}$) is calculated as:(5)$$R_{corr}^{-1}=\frac{\sum_{j=1}^{kl}\left({\vec{x_{j}}\vec{x_{j}}}^{T}\right)}{kl}$$

#### 2.5.4. Matched Filter

Matched filter (MF) is an algorithm used to quantify the abundance of a known spectral signature by partially unmixing the desired endmember from the image and maximizing the intensity while minimizing the remaining signatures [46,47]. The MF algorithm increases the signal-to-noise ratio for the desired endmember while minimizing the background signals by calculating a rejection operator (Equation (6))
(6)$$P=I-UU^{\#}$$
where U represents all signatures in the spectral library acquired a priori while excluding the target endmember, and $U^{\#}$ is the pseudo-inverse of U and is calculated by:(7)$$U^{\#}={\left(U^{T}U\right)}^{-1}U^{T}$$

The rejection operator (Equation (8)) is then used to calculate the filter operator, $a_{i}$, (Equation (9)) used for calculating the abundance via the MF method.
(8)$$\vec{q}={\vec{r_{i}}}^{T}P$$
(9)$$a_{i}=\vec{q}(\vec{x_{i}})$$

### 2.6. Theoretical Sensitivity Analysis Approaches

A theoretical sensitivity analysis (TSA) framework that was previously described [34] was implemented to estimate the performance of various spectral image analysis algorithms for the detection of the Cal 520 signal in excitation-scanning spectral image data. To accomplish this, a region of interest (ROI) containing all spectral components (except for the endmember spectra) was selected from each control sample to perform the TSA. Next, a binary mask corresponding to the selected ROI was created using ImageJ (Equation (6)), and the target endmember was multiplied by a scalar (a) and added to each pixel in the region.
(10)$${\vec{x_{i}}}^{´}=\vec{x_{i}}+a\vec{r_{i}}$$

The value of *a* was varied across a range resulting in a series of images, each with a different level of the target signature added to a selected region of the spectral autofluorescence image. This enabled the creation of a specified region within the image with a known simulated ground truth for the target signature—in this case, Cal 520.

The resulting spectral images were analyzed to produce three curves: a theoretical sensitivity curve (TSC, see example in Figure 2A), a thresholded positive pixel curve (TPPC, Figure 2B), and a receiver operator characteristic (ROC, Figure 2C) curve. The TSC was generated by plotting the unmixed intensity of each pixel in the selected ROI for different levels of the added endmember signal. The black squares represent unmixed pixel intensities, while the red error bars represent the standard deviation in pixel intensities. The mean pixel intensity was also calculated for each level of the added endmember signal and fit to a linear response. The TSC can be used to visualize the linearity of the detected signal response.

The TPPC was generated by plotting the number of pixels within the entire image that were detected as “positive”, as defined by an intensity above a specific threshold—in this example, the threshold = 15 A.U. The TPPC can be used to visualize the number of unmixed “positive” pixels above this fixed threshold value as a function of the endmember signal added to pixels in the ROI. The TPPC also allowed visualization of the detection response slope and the false-positive rate. A more vertical response (increased slope) signifies an increased ability to discriminate negative and positive pixels.

The ROC curve [53] was generated by counting the number of positive pixels that were detected within the ROI as true-positive detections while counting the number of positive pixels counted outside of the ROI as false-positive detections. The number of true-positive and false-positive detections was measured for a range of detection threshold levels at a fixed level of target endmember signal that was added to the ROI. The ROC curve provides an overall visualization of spectral analysis algorithm performance, with ideal target detection represented by an area under the curve equal to 1.

### 2.7. Pixel Threshold Analysis Approach

To improve the reliability of kinetic measurements and to remove bias from non-cell background pixels, the detection threshold can be utilized to allow quantification of only pixels above a minimum detectable limit within a field of view or single-cell region. Utilizing a pixel filtering technique may help to identify subtle changes in time course dynamics that would not be visible when averaging signals from all pixels within a field or large region. These signals, essentially, are hidden or diluted by signals from background pixels or those with weak labeling. To implement the pixel filtering approach, a binary mask was created and used to filter the Cal 520 signal using three different threshold levels: 0 (no threshold), 20 (threshold calculated using the theoretical sensitivity analysis), and 24 (threshold calculated using Otsu, as implemented in ImageJ software [50]). Time-dependent Cal 520 signal levels were quantified for the entire field. In addition, single-cell regions were defined, and Cal 520 signal traces were measured for three different cells using each pixel-filtering threshold level.

## 3. Results and Discussion

Hyperspectral imaging approaches have been applied to microscope systems using a variety of spectral technologies: LCTFs [18,19], AOTFs [13,20], TFTFs [22], gratings [36], prisms [37], interferometers [54], spectral snapshot cameras [55], and light-emitting diode (LED) arrays [56,57]. Each technology has associated advantages and disadvantages, and in some cases, may be implemented in more than one way, such as for scanning of the fluorescence emission or excitation spectrum. The focus of this study was to utilize HSI microscopy for a particularly challenging application—measurement of time-dependent fluorescence in the midst of cellular autofluorescence for dynamic, live-cell, cell signaling experiments. The spectral imaging requirements for this application include the need for:High sensitivity—sufficient to detect subtle differences in the Cal 520 signal due to temporal or spatial changes in Ca^2+^ concentration.High speed—sufficient to detect dynamic changes in the Cal 520 signal on the order of 20–30 s consistent with slower Ca^2+^ signals or waves.High photon efficiency—necessary to minimize photobleaching effects during time-lapse experiments.High spatial resolution—sufficient to allow localization of Ca^2+^ signals to spatial regions within the cell, such as near the cell membrane, near the nucleus, etc.Moderate field of view—sufficient to allow simultaneous imaging of multiple (~20–100) cells.Moderate spectral resolution—sufficient to allow discrimination of spectral peaks from Cal 520, NucBlue, and cellular autofluorescence—a spectral resolution of 10–20 nm is likely sufficient for this study as Cal 520 and NucBlue excitation wavelength peaks are ~150 nm in separation, and autofluorescence presents a broad intensity across this excitation spectral range.A moderate and compromised spectral bandwidth—sufficiently narrow to achieve the needed spectral resolution (the TFTF system provides a bandwidth of ~18–20 nm), but sufficiently broad as to provide a strong excitation intensity for detecting weak signals.A clean spectral band with high out-of-band rejection—sufficient to prevent excitation-emission spectral cross-talk and to minimize cross-talk between spectral bands.

The excitation-scanning HSI microscope system and corresponding acquisition settings were selected for this study to satisfy the range of spectral imaging requirements described above. In specific, the use of a TFTF array provides ~95% optical transmission, a bandwidth of between 18 and 20 nm depending on center wavelength, and a clean spectral band with out-of-band rejection of OD 5+. The excitation-scanning approach provides high overall photon efficiency for the system and helps to minimize photobleaching. These capabilities could be further enhanced by the development of very-high-speed rotation mechanisms to allow rapid tuning of the TFTF array, which could reduce the temporal sampling speed to 1–2 s per spectral band and allow the sampling of rapid Ca^2+^ transient signals.

### 3.1. Experimental Results and Discussion

Agonist-induced Ca^2+^ responses were measured using excitation-scanning HSI and analysis approaches. To provide the a priori information required for linear unmixing, as well as other spectral analysis approaches, images of single-label control samples were acquired for NucBlue (Figure 3A), unlabeled cells for cellular autofluorescence (Figure 3B), and Cal 520 (Figure 3C). The magnitude of signals within spectral images was visualized by summing all spectral bands. A region was then selected in each single-label image that corresponded to an area of high signal intensity. This region was transferred to the corresponding spectral image stack, and the pixel-averaged spectrum was extracted, yielding the measured reference spectrum (also known as the pure endmember spectrum). Pure spectra were combined into a spectral library (Figure 3D) for use with spectral analysis algorithms. Spectra were also validated against manufacturer-supplied and previously reported excitation spectra, as measured using a spectrofluorometer (Figure 3E). In general, there was high agreement between the spectral shape of the measured pure spectra and previously reported spectra.

Fluorescence excitation-scanning HSI data of Ca^2+^ signaling in HASMCs were then analyzed to detect each endmember. For visualization in a print format, single time points have been selected for viewing time-lapse microscopy data (Figure 4). The full time-lapse image data set for this study is available in Appendix A. Results from non-negatively constrained linear unmixing (LU) demonstrate the ability to separate the Ca^2+^ label signal (Cal 520) from the NucBlue and Autofluorescence. Results indicate that all three signals are identifiable and that the residual root mean square (RMS) (Figure 4) error is less than any of the unmixed endmembers. Increased Ca^2+^ signaling activity is clearly visible at time point 340 s after the addition of 50 µM carbachol, and dynamics can be seen in Appendix A.

Four spectral analysis algorithms—LU, SAM, CEM, and MF—were utilized to evaluate the ability to separate and identify endmember signals in fluorescence excitation-scanning time-lapse HSI data (Figure 5). While these algorithms have been previously used for the analysis of remote sensing HSI data, only the LU algorithm has been applied to fluorescence excitation-scanning HSI microscopy data. Hence, it is important to evaluate the performance of these algorithms with respect to the ability to detect Cal 520 and NucBlue signals and the ability to accurately differentiate these signals from confounding cellular autofluorescence. This is especially important when considering that signal levels and the mixture of spectral signatures will dynamically change for time-lapse live-cell imaging studies.

In a side-by-side comparison, all four algorithms were able to identify dynamic changes in Cal 520 due to changes in the Ca^2+^ concentration. However, the different algorithms presented different abilities to separate endmember signals. For example, SAM is a spectral angle measure and not an unmixing algorithm and, as such, provided spectral angle measurements that, when viewed as images, displayed contributions from endmembers other than the target endmember. Most notable is that the AF signal is visible in the SAM channel for Cal 520. All the other three analysis algorithms (CEM, LU, and MF) are subpixel analysis techniques and provide endmember images with visibly lower contributions from endmembers other than the target endmember. However, subtle analysis artifacts can still be visualized, especially when viewing time-dependent analyzed image data (Appendix A). For example, time-dependent changes in NucBlue can be seen, which are likely artifactual. The nuclei signal should be relatively static over the time span of these experiments, and rapid changes in the nuclear signal are likely due to the Cal 520 signal being incorrectly identified as NucBlue. This incorrect signal assignment is more apparent in CEM than in LU and MF and is also present in the AF channel.

An important consideration when analyzing HSI fluorescence microscopy data, especially from dynamic live cell studies such as this, is that generation of an image data set with a known ground truth is experimentally prohibitive. For cases of fixed cells or fluorescently labeled microspheres, it may be possible to experimentally generate a sample with a known ground truth through very precise fluorescent label titration so as to achieve a specified labeling density. However, for live cell samples, knowledge of the exact labeling density (the number of labels per area or per volume) is prohibitively difficult to ascertain due to an unequal uptake of fluorescent labels by the cell, unequal distributions of cell label binding sites, non-specific binding of the label to other sites, variations in cellular autofluorescence, variations in cell geometry (a typical “fried-egg” shaped cell will be thicker near the nuclear region and very thin near the periphery), and important variations in these parameters over time due to both normal and induced changes in cell physiology. To address these concerns, we have previously utilized a “theoretical sensitivity analysis” framework for probing the sensitivity of a spectral imaging assay, with a corresponding analysis algorithm, to varying theoretical levels of signals that are artificially added post-acquisition to a spectral image dataset [34]. In this study (Section 3.2 and Section 3.3), we extend the theoretical sensitivity analysis framework to differentiate between pixels that are above a minimum detectable limit threshold and pixels that are below the detectable limit and may be considered background.

### 3.2. Theoretical Sensitivity Analysis Results and Discussion

In microscopy, it is often prohibitively difficult to establish a known ground truth for a specific location in a cell signaling experiment. To assess spectral image analysis performance, a theoretical sensitivity framework may be used to estimate detection sensitivity and specificity. This is advantageous as the contributions due to cellular autofluorescence are unknown and variable. Here, we evaluate the performance of four conventional spectral analysis algorithms (LU, SAM, CEM, and MF) for identifying a target endmember in the midst of other endmembers and confounding AF in fluorescence excitation-scanning HSI data (Figure 6).

The TSC was used to visualize the analyzed intensity of each pixel within a region as a function of the amount of target endmember added. LU (Figure 6A) and MF (Figure 6J) analyses provided a linear response (denoted by the black line) to increasing target endmembers, while SAM (Figure 6D) and CEM (Figure 6G) responses were nonlinear. By inspecting the standard deviation (red error bars), it can be seen that the standard deviation for LU and MF were insensitive to the amount of target endmember added, while SAM showed a decreasing standard deviation as a function of increasing endmember added, and CEM provided an increasing standard deviation with each target endmember added.

The threshold positive pixel curve (TPPC) was used to visualize the ability to detect positive pixels in the image when given a specified detection threshold. Specifically, the TPPC provides the ability to visualize both the false-positive level (the baseline number of positive pixels detected when no target endmember signal is added) as well as the slope of the transition from pixels being detected as negative to being detected as positive. LU and MF algorithms (Figure 6B,K) displayed a more vertical slope of the TPPC, indicating the ability to detect positive pixels once the amount of the target endmember was at or above the detection threshold. For SAM (Figure 6E), pixels with angles less than the detection threshold (in this case, 0.98 radians) were defined as positive, as the SAM algorithm calculates the spectral angle in radians where a small spectral angle indicates a high similarity between endmember and unknown pixel spectra. For SAM, 147,267 pixels (29.5% of total pixels) were detected as positive (below the 0.98 radian threshold) prior to adding any Cal 520 signal. This indicates that the AF signature was sufficiently similar to Cal 520 in these pixels to produce a false-positive detection. A 29.5% false-positive detection rate is unacceptable for sensitive signal detection. Hence, if SAM were to be utilized, a more stringent threshold should be selected at the expense of reduced sensitivity for detecting true-positive pixels (see Appendix A for alternative SAM classification threshold that results in decreased false positives, but also inability to detect weak true positive signals).

The ROC curve was used to visualize the overall performance of pixel classification based on results from each analysis algorithm. The area under the ROC curve is typically the most valuable indicator of detection performance, although the shape of the ROC curve may also be informative. All four algorithms produced different ROC curve shapes, with LU (Figure 6C) presenting the best result with an area under the curve (AUC) of 0.963,and SAM, CEM, and MF (Figure 6F,I,L), with areas of 0.570, 0.872, and 0.957, respectively. This signifies that when performing a sensitivity analysis based on the ROC curve, LU and MF provide similar performance that is much higher than SAM or somewhat higher than CEM.

### 3.3. Pixel Threshold Analysis Results and Discussion

Biological samples frequently have higher variability than is encountered in traditional satellite or aerial remote sensing applications. In addition, cellular autofluorescence often confounds measurements of target signatures. We have previously shown that a theoretical sensitivity analysis framework can be used to provide an estimate of the images with varying autofluorescence. A minimum detectable limit (the detection threshold) is evaluated in theoretical sensitivity analysis, and this limit can be further used to identify pixels that have a detectable signal level and those that do not (a signal may be present but is not above the detection threshold). This is advantageous, as traditional cell signaling assays often quantify the signal level for the entire field of view or, at best, specify a region of interest, such as one or several cells. However, when specifying the whole field of view or large regions, it is likely that an error will be present due to non-cell background pixels. In addition, when analyzing data from specific, single-cell regions, there may be pixels within a region that present low or, in practice, unusable levels of fluorescent labels. These weakly labeled pixels may bias extracted time-dependent kinetic measurements.

The Cal 520 signal was filtered using three different threshold levels: 0 (no threshold, Figure 7A), 20 (threshold calculated using the theoretical sensitivity analysis, Figure 7B), and 24 (threshold calculated using Otsu, Figure 7C). The Cal 520 signal levels were quantified for the entire field (Figure 7D). Each pixel filtering threshold level was used to measure traces of the Cal 520 signal for defined single-cell regions (Figure 7E–G). We observed different signal trace kinetics for the entire field (ensemble average response) and for each cell selected, although all cells displayed increased signaling activity after treatment with 50 µM carbachol. These observations can be seen in Appendix A. The application of pixel filtering through the binary mask resulted in the identification of subtler Cal 520 signal variations for the entire field, as well as for cells 1 and 2. The results of pixel thresholding were least pronounced for cell 3, which appeared to be very well labeled (all pixels above the minimum detectable limit threshold). Hence, a pixel thresholding technique such as this may be helpful in identifying subtle changes in Ca^2+^ of other cell signaling agent dynamics, given the caveat that pixel filtering also may provide an additional analysis step that could introduce signal measurement bias or artifacts if used inappropriately.

To further evaluate the effects of selecting pixels above a minimum detectable limit for measurement of Cal 520 signal dynamics, additional experiments were conducted utilizing HASMCs treated with either 50 μM carbochol (Figure 8) or 50 μM histamine (Figure 9). Hyperspectral images were analyzed using the MF algorithm and thresholded as described above (Figure 7) to retain only pixels that were above a defined minimum detectable limit. Single-cell ROIs were selected, and region-averaged signal traces were extracted for four representative cells. A comparison of time traces before and after pixel thresholding revealed that subtle variations in single-cell Cal 520 signals could be visualized in the pixel-thresholded image sets (compare cell 1 time traces for non-thresholded, Figure 8C, and thresholded, Figure 8F plots). These results indicate that there are likely transient signal variations within cell 1, as well as other cells, that may be masked or altered by pixels below the minimum detectable limit and which become further pronounced with pixel-thresholding techniques.

The effects of pixel thresholding were also investigated using cells treated with an alternative agonist, 50 μM histamine (Figure 9). In this experiment, pixel thresholding resulted in changes in Cal 520 traces for several of the single cells investigated. For example, cell 1 displayed increased fluctuations in Cal 520 signal levels prior to treatment with pixel thresholding (Figure 9F) when compared to non-thresholded images (Figure 9C). These results indicate that there were perhaps time-dependent fluctuations in a subset of pixels within cell 1 but that when averaging all pixels within the cell, these fluctuations were damped by a large number of low-intensity pixels. In contrast, when averaging only pixels above the minimum detectable limit threshold, these fluctuations were more pronounced. In addition, pixel thresholding appeared to bring the baseline value for cells 2–4 to a more consistent value, indicating that perhaps in non-thresholded data, these cells had variable amounts of low-intensity pixels (below the minimum detectable limit) that affected the whole-cell Cal 520 signal level. Taken together, these data indicate that a single-cell ROI may likely contain a mixture of low-intensity pixels that are below a minimum detectable limit for Cal 520 and high-intensity pixels that are above the limit. In addition, changes in intensity may not occur uniformly. In the case of cell 1, there are likely high-intensity pixels that undergo large fluctuations in intensity as well as low-intensity pixels that undergo minimal fluctuations in intensity. Hence, when measuring single-cell Cal 520 time traces from analyzed HSI data, the trend and accuracy of results are likely dependent on both the ability to perform accurate cell segmentation as well as on whether all pixels within a cell ROI are considered or only pixels above a defined minimum detectable limit.

### 3.4. Considerations for Live-Cell HSI Studies

In this study, fluorescence excitation-scanning HSI microscopy was utilized to study dynamic Ca^2+^ signals in HASMC preparations using time-lapse imaging. Analysis of dynamic signals in live cell samples presents an especially unique set of challenges due to requirements for temporal, spectral, and spatial sampling; effects of photobleaching and cell movement; and confounding factors such as autofluorescence. Detailed descriptions of these requirements for live cell imaging are provided at the beginning of Section 3. The results from this study have indicated that excitation-scanning HSI can provide an effective approach for separating desired fluorescence labels, such as Cal 520, from other labels as well as competing autofluorescence. Of the four spectral analysis algorithms evaluated, LU and MF provided similar results that were deemed to be overall more accurate than SAM or CEM algorithms. MF provided a reduced computational burden and was utilized for further study of time-course measurements from specific cells of interest. The use of pixel-wise thresholding provided an additional ability to improve the reliability of time-course measurements by rejecting pixels below a specified minimum detectable limit.

The advantages of this combined excitation-scanning hardware and custom analytical approach include the ability to measure time-dependent fluorescence image data and to discriminate among fluorescent labels and cellular autofluorescence—in general, the approach is effective. However, there are several considerations and limitations that should be noted. First, the current hardware embodiment of the excitation-scanning HSI microscope is limited in terms of the wavelength switch time by the need to mechanically rotate between adjacent TFTFs housed in a filter wheel and the need to tilt the TFTF filter wheel so as to tune the center wavelength band. There is much potential to improve the wavelength tuning speed through the revision of the optomechanical tuning assembly and the potential use of high-speed piezoelectric rotation stages or galvanometers.

Second, when configuring an HSI microscope system for a particular assay, it is important to match the spectral scanning parameters to the fluorescent labels or, alternatively, to select appropriate fluorescent labels that match the spectral scanning capabilities of the microscope system. In this study, the Cal 520 and NucBlue excitation peak wavelengths were separated by ~150 nm (see Figure 3), with a broad cellular autofluorescence covering the entire spectral scan range. While the excitation spectral scan range of 360–480 nm was sufficient to capture the full signature of NucBlue, it was only sufficient to capture part of the spectral signature of Cal 520, with the peak excitation wavelength reported by the manufacturer to occur at ~495 nm. If available, the selection of an alternative Ca^2+^ label with a lower excitation peak—for example, at 425 nm—would have allowed more of the spectral signature to be detected using the current scan range and may have provided an improved ability to separate signals from each fluorescent label. Alternatively, the use of such an alternative fluorescent label could enable a reduced excitation-scanning spectral range, resulting in an increase in overall spectral image acquisition time and enabling either reduced photobleaching or improved temporal sampling.

Third, the approach of fluorescence excitation-scanning HSI carries the potential to generate time-dependent spectral artifacts if high-speed changes in fluorescent label distribution or intensity occur between the onset and the completion of a spectral scan. These artifacts could affect the accuracy of any resulting spectral image analysis algorithms. However, it should be noted that alternative push–broom or whisk–broom spectral scanning techniques would also present the capacity for a motion-induced artifact, but in this case, the artifact would not affect the spectral domain, but would instead manifest as a spatial artifact, as changes to label distribution or intensity could occur between the scanning of one pixel and an adjacent pixel. Snapshot spectral imaging may present an alternative approach to mitigate these motion artifacts but presents a trade-off in providing reduced spatial resolution/reduced field of view, as well as reduced spectral resolution and the inability to scan continuously across a range of spectral bands. It is important to consider these potential limitations and potential sources of artifacts when selecting a system and settings for HSI fluorescence microscopy.

## 4. Conclusions

Hyperspectral imaging technologies have undergone rapid development, from initial applications in remote sensing to a wide range of applications, including biomedical imaging and fluorescence microscopy. However, there is still much potential to improve both HSI hardware and software for use in live cell fluorescence microscopy. New approaches are needed that provide improved temporal sampling and reduced photobleaching. The excitation-scanning HSI system utilized in this study has been previously demonstrated to provide improved acquisition sensitivity and speeds in comparison to a similarly configured emission-scanning HSI system. However, these prior studies were performed in non-dynamic experimental conditions. Here, we have demonstrated the use of excitation-scanning HSI for live cell dynamic Ca^2+^ imaging. We have also evaluated four analysis algorithms that have been previously used in the remote sensing field (LU, SAM, CE, and MF) for the ability to detect and discriminate fluorescence excitation signatures. All three subpixel analysis techniques (CEM, LU, and MF) provided a high ability to quantify the Cal 520 signal. As seen in Figure 6, LU and MF provided a linear response to increasing Cal 520 signal and a steep slope of the TPPC, indicating that both are effective algorithms for quantifying Cal 520. At a threshold value of 15, LU resulted in 0 false-positive detections, while MF resulted in 127 (out of a total of 498,436 pixels within the image); hence, LU provided slightly improved false-positive rejection. The classification analysis algorithm (SAM) was able to identify regions of the Cal 520 signal but displayed a nonlinear dependence endmember signal. Hence, SAM could be useful for identifying regions within an image but was not suitable for quantifying Cal 520 signal over time.

To improve the ability to extract dynamic Ca^2+^ signals from small ROIs, the theoretical sensitivity framework was extended to allow the estimation of a minimum detectable limit for use in pixel filtering. The results indicated that pixel filtering could be used to remove bias from unlabeled or weakly labeled pixels, especially when measuring bulk signals from an entire field of view. When pixel filtering was applied to whole-field time-lapse data, the resultant Cal 520 time course included subtle kinetic features that were not present in unfiltered data—features that were likely washed out by averaging with background pixels. This approach could be further extended to a range of dynamic cell signaling experiments and may also be modified to allow for the use of autofluorescence signals as either fiducial or metabolic markers [35]. Hence, the use of excitation-scanning HSI with suitable spectral analysis algorithms and pixel filtering may provide a mechanism for measuring subtle kinetic events in cell signaling and other live cell dynamic assays.

## Figures and Tables

**Figure 1 bioengineering-10-00642-f001:**
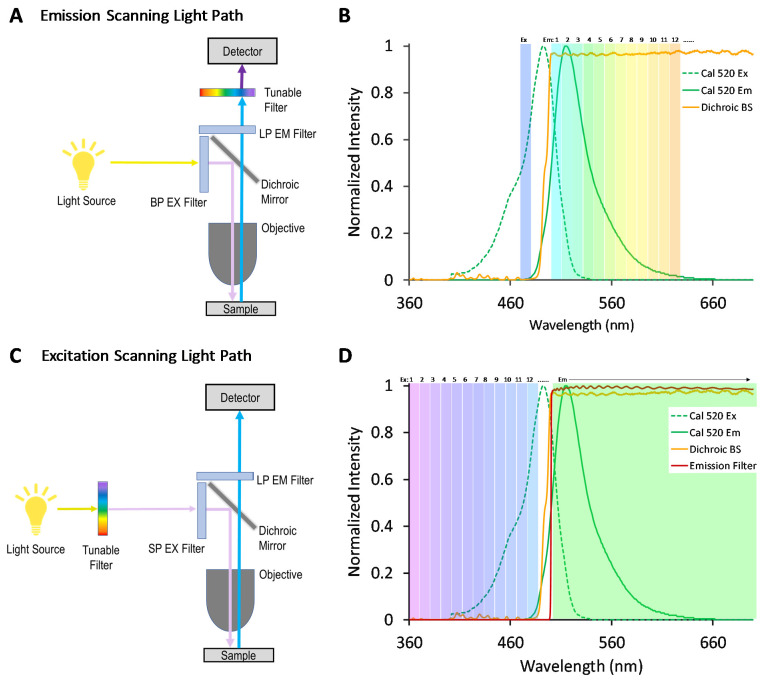
Light path schematics and corresponding spectral scan range illustrations for emission-scanning and excitation-scanning HSI microscope systems. (**A**) A typical emission-scanning HSI microscope system utilizes one or several narrow bandwidths of illumination for excitation while acquiring fluorescence emission over many narrow spectral bands using a tunable filter or other wavelength sampling device (prism, grating, etc.). (**B**) An illustration of the spectral scan range for an emission-scanning HSI system. The fluorescence excitation and emission spectra of a hypothetical fluorescence label are shown as dotted and solid green lines. Excitation is provided at one narrow band, illustrated by the solid green bar placed at the peak excitation wavelength. Fluorescence emission is sampled across many narrow bands, illustrated by the many individual bars across the fluorescence emission spectrum. (**C**) The excitation-scanning HSI microscope system utilizes a tunable filter to sequentially select between many narrow excitation wavelength bands while acquiring fluorescence emission using a broad-band or long-pass emission filter (LP EM Filter). (**D**) An illustration of the spectral scan range for an excitation-scanning HSI system. The fluorescence excitation spectrum is sequentially sampled using many narrow excitation bands, as indicated by narrow bars placed over the excitation spectrum. The emitted fluorescence is detected in bulk using a broad-band or long-pass emission filter.

**Figure 2 bioengineering-10-00642-f002:**
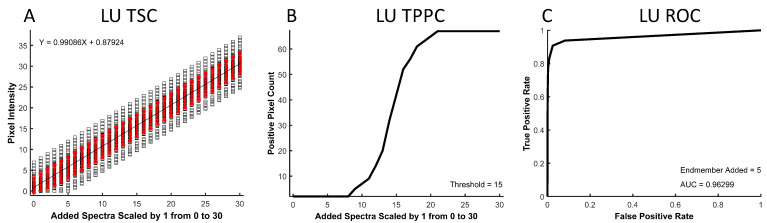
An example theoretical sensitivity analysis, as performed for non-negatively constrained linear unmixing (LU). (**A**) The theoretical sensitivity curve (TSC) demonstrates a linear response of the LU algorithm to varying levels of the added target endmember, Cal 520. (**B**) The thresholded positive pixel curve (TPPC) indicates a sharp slope in detection accuracy for a threshold of 15 (unmixed intensity units). Importantly, using the threshold of 15, no false-positive pixels were detected when the 0 endmember signal was added. (**C**) The receiver operator characteristic (ROC) curve demonstrates a high predicted performance for LU for this application of Cal 520 detection, with an area under the curve (AUC) of 0.96. The TSC analysis was performed for all 4 spectral analysis algorithms, as described in the Results.

**Figure 3 bioengineering-10-00642-f003:**
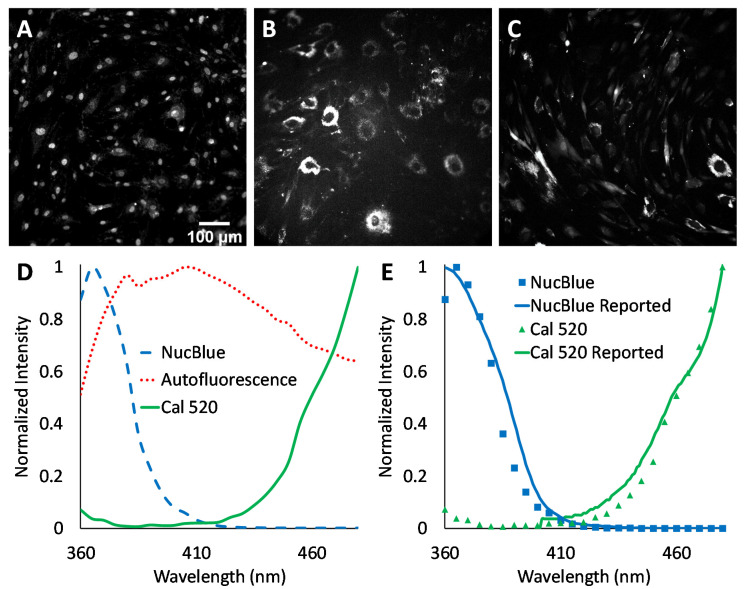
Spectral image data from three single-label control samples were analyzed to construct a spectral library. (**A**) Representation of HSI image data from HASMCs labeled with the nuclear label, NucBlue. For visualization purposes, all wavelength bands have been summed to represent a total or summed fluorescence intensity image, and the intensity range linearly adjusted from 0 to 6300 A.U. for display. (**B**) A summed fluorescence intensity representation of HSI image data from a separate sample of unlabeled HASMCs with intensity range linearly adjusted from 0 to 315 A.U. for display. (**C**) A summed fluorescence intensity representation of HSI image data from a separate sample of HASMCs labeled with Cal 520 with intensity range linearly adjusted from 0 to 550 A.U. (**D**) A spectral library was generated by selecting a region of high signal intensity within each of the single-label control spectral images (**A**–**C**), extracting the pixel-averaged spectrum, and normalizing to a peak value of unity. (**E**) Comparison of measured spectra for NucBlue (blue squares) and Cal 520 (green triangles) to reported spectra. The Cal 520 spectrum was supplied from AAT Bioquest, while the NucBlue spectrum was approximated as that of DAPI and obtained using the Semrock Searchlight spectral plotting tool [58].

**Figure 4 bioengineering-10-00642-f004:**
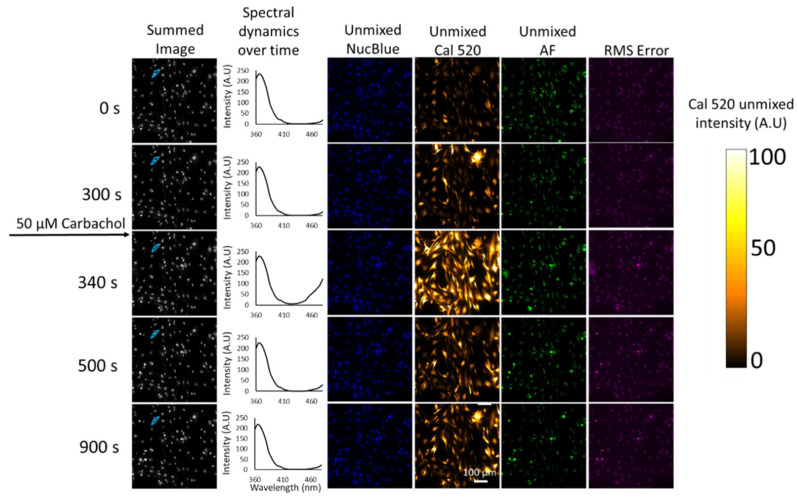
Linear spectral unmixing of spectral images from Cal 520 labeled HASMCs. All wavelength bands were summed to show the total intensity of raw spectral images (column 1) at different time points. A region of interest corresponding to a single cell was selected to illustrate the change in excitation spectrum over time (Column 2). The primary contributors to the mixed spectra shown in Column 2 were NucBlue (excitation peak at 360 nm) and Cal 520 (excitation peak at 520 nm). Note that the excitation peak of Cal 520 is beyond the scan range used for the current study, and hence the highest wavelength response for Cal 520 occurs at the last wavelength scanned of 480 nm—see Figure 3E for a comparison of measured and manufacturer excitation spectra. Note that the Cal 520 contribution is increased at time point 340 s, which corresponds to 40 s after agonist addition and a strong Ca^2+^ release response visualized in the unmixed Cal 520 images (Column 4). Columns 3–5 show false-colored unmixed endmember images of NucBlue, Cal 520, and AF signals. Column 6 shows the RMS error associated with linear spectral unmixing. The color bar at the right represents the color look-up table used to visualize Ca^2+^ signal intensity in Column 4.

**Figure 5 bioengineering-10-00642-f005:**
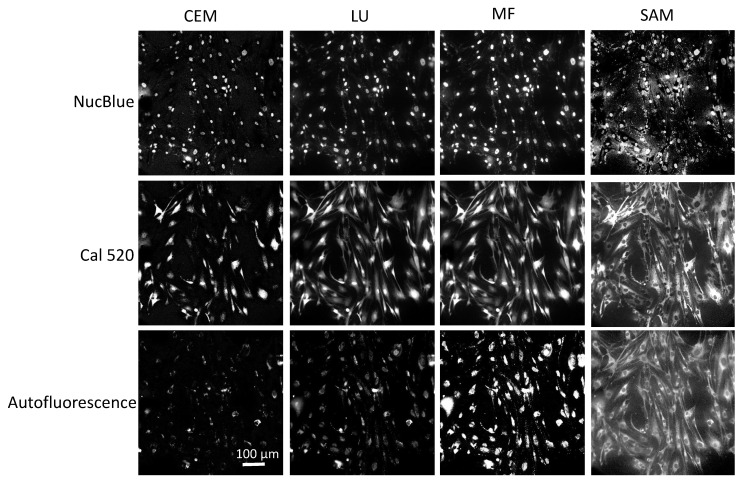
A comparison of 4 common spectral analysis algorithms for identifying endmembers in excitation-scanning spectral image data. Analyzed images were displayed using a greyscale look-up table for each endmember: NucBlue, Cal 520, and autofluorescence (AF). Images were analyzed using constrained energy minimization (CEM), linear unmixing (LU), matched filtering (MF), and spectral angle mapper (SAM). A video showing the full timelapse data set can be viewed in Appendix A. The same look-up tables were used for both Figure 4 and Figure 5 to permit comparison.

**Figure 6 bioengineering-10-00642-f006:**
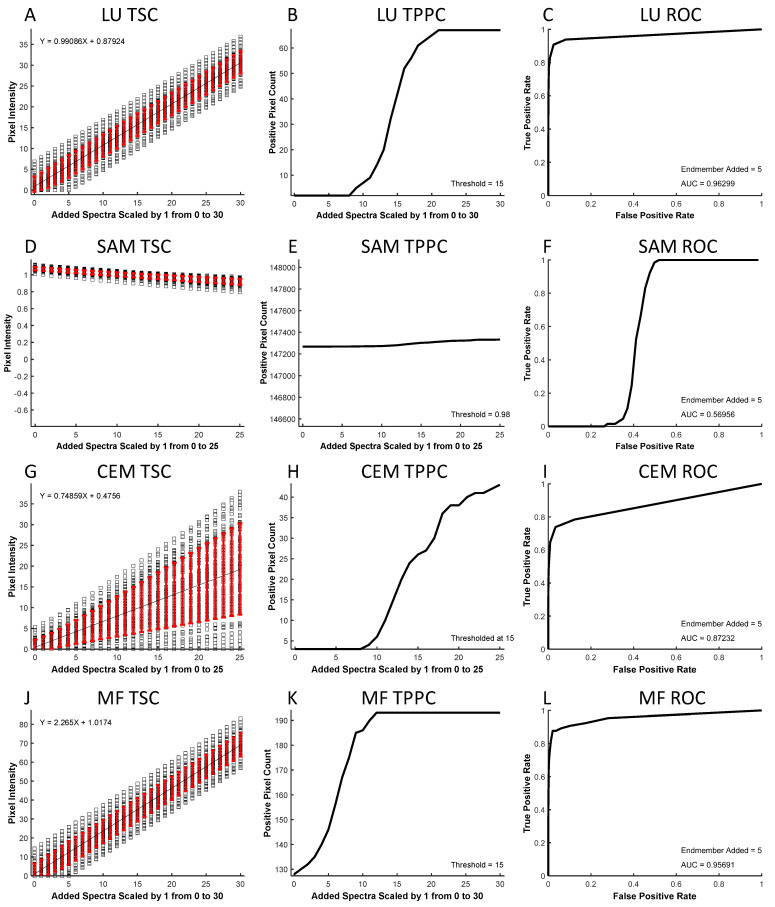
A theoretical sensitivity analysis was applied using 4 spectral analysis algorithms for identifying fluorescence excitation-scanning signals. Three curves were calculated for each analysis algorithm: the Theoretical Sensitivity Curve (TSC), Thresholded Positive Pixel Curve (TPPC), and Receiver Operator Characteristic (ROC) curve. (**A**–**C**) TSC, TPPC, and ROC for linear unmixing (LU); (**D**–**F**) TSC, TPPC, and ROC for spectral angle mapper (SAM); (**G**–**I**) TSC, TPPC, and ROC for constrained energy minimization (CEM); (**J**–**L**) TSC, TPPC, and ROC for matched filter (MF). TSC and TPPC plots were generated by adding the spectrum of the target endmember, Cal 520, to a specified region of interest (ROI) in set amounts (increments of 1 were 0–30, 0–25, 0–25, and 0–30 for LU, SAM, CEM, and MF, respectively). ROC scale factors were 5 for all algorithms. Threshold values for TPPC were 15 for LU, 0.98 radians for SAM, and 15 for both CEM and MF. ROC threshold values were varied from 0 to 50 in increments of 1 for LU, CEM, and MF. ROC threshold values for SAM varied from 0 to 1.8 radians. ROC performance, as described by area under the curve (AUC), was 0.963 for LU, 0.570 for SAM, 0.872 for CEM, and 0.957 for MF.

**Figure 7 bioengineering-10-00642-f007:**
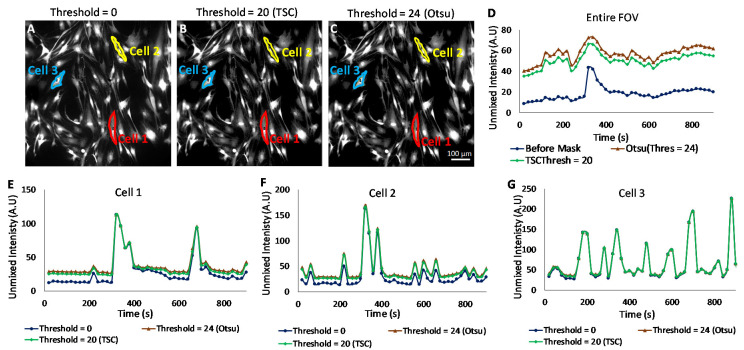
Effects of pixel filtering on time-lapse Ca^2+^ image data acquired using excitation-scanning spectral imaging microscopy and matched filter (MF) analysis. Thresholds for pixel filtering were determined using a theoretical sensitivity analysis and a standard Otsu thresholding algorithm available in ImageJ. (**A**) A representative endmember image of Cal 520 at time point 40 s after treatment obtained using the MF algorithm. The Cal 520 image from panel (**A**) was also processed using pixel filtering to remove low-intensity (below detection limit) pixels using a threshold estimated from the theoretical sensitivity analysis ((**B**), threshold = 20) and from Otsu ((C), threshold = 24). (**D**) The field of view averaged time course for Cal 520 signal as estimated from the original and pixel-filtered image sets. Agonist-induced Ca^2+^ signals in three different cells (cells 1, 2, and 3 shown as red, yellow, and blue regions on image panels (**A**–**C**)) were measured using the original and pixel-filtered image sets ((**E**)—(cell1), (**F**)—(cell2), and (**G**)—(cell3)).

**Figure 8 bioengineering-10-00642-f008:**
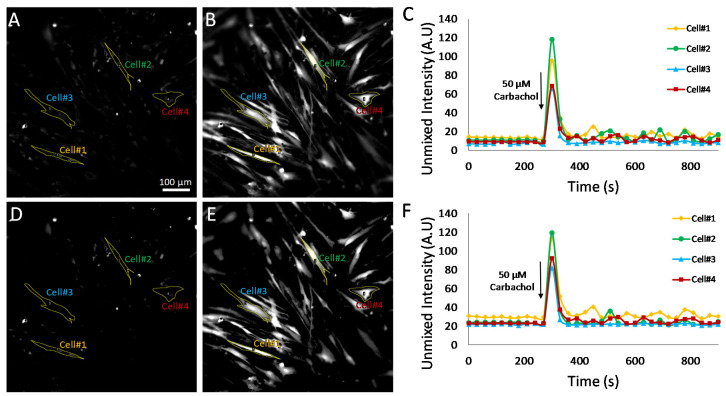
Effects of pixel filtering were further evaluated using time-lapse excitation-scanning HSI data of Cal 520 and NucBlue labeled HASMCs to identify dynamic Ca^2+^ signals. Hyperspectral images were analyzed using a matched filter (MF) algorithm, and thresholds for pixel filtering were set to a value of 20, corresponding to the theoretical sensitivity analysis identified threshold used in Figure 7B. (**A**) The initial time point Cal 520 endmember image (time point 0 s) corresponding to baseline Cal 520 fluorescence. Outlines for single-cell regions of interest (ROIs) are shown in yellow, and the cell number is indicated by colored text. A video showing all time points is provided in Appendix A. (**B**) The Cal 520 endmember image at a time point of 300 s, corresponding to a time point immediately after addition of 50 μM carbochol. A high-intensity response is seen in most cells. (**C**) The corresponding time trace for Cal 520 intensity for each of the single-cell ROIs. (**D**) The Cal 520 endmember image at time point 0 after applying a pixel threshold of 20. (**E**) The Cal 520 image at time point 300 after applying a pixel threshold of 20. A video showing all time points after pixel thresholding is provided in Appendix A. (**F**) The corresponding time trace for pixel-thresholded Cal 520 intensity for each of the single-cell ROIs. Periodic oscillations in Cal 520 intensity can be seen in cell 1 in the pixel-thresholded image data that are not visible in the non-thresholded data.

**Figure 9 bioengineering-10-00642-f009:**
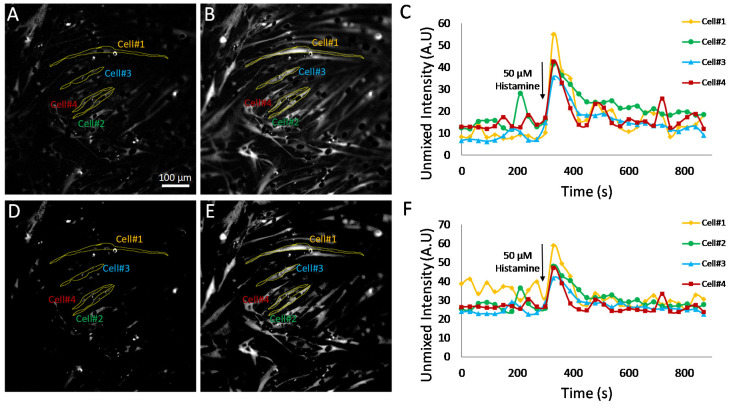
Effects of pixel filtering were also evaluated in hyperspectral images of HASMCs treated with 50 μM histamine. Hyperspectral images were acquired and analyzed identically to images shown in Figure 8. (**A**) The initial time point Cal 520 endmember image (time point 0 s), corresponding to baseline Cal 520 fluorescence. Outlines for single-cell regions of interest (ROIs) are shown in yellow, and the cell number is indicated by colored text. A video showing all time points is provided in Appendix A. (**B**) The Cal 520 endmember image at a time point of 330 s, corresponding to a time point immediately after addition of 50 μM carbochol. A high-intensity response is seen in most cells. (**C**) The corresponding time trace for Cal 520 intensity for each of the single-cell ROIs. (**D**) The Cal 520 endmember image at time point 0 after applying a pixel threshold of 20. (**E**) The Cal 520 image at time point 330 after applying a pixel threshold of 20. A video showing all time points after pixel thresholding is provided in Appendix A. (**F**) The corresponding time trace for pixel-thresholded Cal 520 intensity for each of the single-cell ROIs. Fluctuations in Cal 520 signal prior to treatment can be seen in cell 1 in the pixel-thresholded image data that are not present in the non-thresholded data.

**Table 1 bioengineering-10-00642-t001:** Experimental spectral microscope acquisition parameters.

Microscope	Light Source	Objective	Detector
TE2000-U inverted epifluorescence widefield microscope (Nikon Instruments)	Titan 300 Xe arc lamp (Sunoptics Surgical) TFTF array: scan range 360–480 nm in 5 nm increments	Plan Apo 20X, N/A 0.75 (Nikon Instruments)	Prime 95B sCMOS camera (Teledyne Photometrics) 30 ms exposure time 0 averaging 2 × 2 binning 0 gain or offset 200 MHz readout speed Overall imaging speed = ~15 s/spectral image stack

## Data Availability

Data utilized in this manuscript are available at https://www.southalabama.edu/centers/bioimaging/resources.html accessed on 22 April 2023.

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
