# Peer review of "Comparing Performance of Spectral Image Analysis Approaches for Detection of Cellular Signals in Time-Lapse Hyperspectral Imaging Fluorescence Excitation-Scanning Microscopy"

_bioengineering, 2023, doi:10.3390/bioengineering10060642_

Round 1

Reviewer 1 Report

Authors have taken carefully check the excitation components for the fluorescent emitted imaging, which are very important process to observe the imaging. It will be acceptable manuscript for this journal.

Author Response

We would like to thank the reviewer for their thoughtful review of the manuscript and encouraging comments.

Reviewer 2 Report

This paper compares the performance of common spectral analysis algorithms for use in kinetic cell signaling experiments. This may be helpful for separation of the fluorescent label of interest from cellular autofluorescence. I think the topic of this paper is interesting. Here are some of the issues with this paper that I would like to point out:

1. The abstract of this article can be more streamlined.

2. Section Introduction: more exhaustive summary of existing methods, and the disadvantages of these methods, as well as the advantage of this proposed method, are expected to be elaborated. For example, hyperspectral imaging has been widely used in different fields:

*Diagnosis of Cholangiocarcinoma from Microscopic Hyperspectral Pathological Dataset by Deep Convolution Neural Networks, Methods, 202: 22-30, 2022.

*Identification of Melanoma from Hyperspectral Pathology Image Using 3D Convolutional Networks. IEEE Transactions on Medical Imaging, 2021, 40(1): 218-227.

3. Figure 1: please compare Fig. 1A and Fig. 1C in detail.

4. Table 1: please list the imaging speed as it is important for fluorescence imaging.

5. I think this paper lacks the discussion of spectral imaging requirements in context of fluorescence applications. For example, what are requirements for fluorescence imaging (spectral resolution, bandwidth, speed) and to perform successful unmix? What would enhance these techniques?

6. Section 2.3 Hyperspectral Image Data Pre-processing: please explain in detail how to normalize the wavelength data.

7. Section 2.4 Reference Spectra and Spectral Library: please explain how to get the pixel-averaged spectrum.

8. Figure 5: I think the results should be compared with the ground truth. I think the paper could be stronger if the performance comparisons have been clearly addressed.

9. As the spectral analysis algorithms have been widely used, please explain what is new in the analysis method used in this paper.

10. A more comprehensive section of discussion is needed to address lots of issues, such as the advantages and disadvantages of the proposed method.

Author Response

This paper compares the performance of common spectral analysis algorithms for use in kinetic cell signaling experiments. This may be helpful for separation of the fluorescent label of interest from cellular autofluorescence. I think the topic of this paper is interesting. Here are some of the issues with this paper that I would like to point out:

<Response> We would like to thank the reviewer for their detailed review and critique of our manuscript.  Below, please see responses to each of the individual comments.

  1. The abstract of this article can be more streamlined.

<Response> Thank-you for the helpful suggestion.  We have now edited the abstract to be more concise and direct.

  1. Section Introduction: more exhaustive summary of existing methods, and the disadvantages of these methods, as well as the advantage of this proposed method, are expected to be elaborated. For example, hyperspectral imaging has been widely used in different fields:

*Diagnosis of Cholangiocarcinoma from Microscopic Hyperspectral Pathological Dataset by Deep Convolution Neural Networks, Methods, 202: 22-30, 2022.

*Identification of Melanoma from Hyperspectral Pathology Image Using 3D Convolutional Networks. IEEE Transactions on Medical Imaging, 2021, 40(1): 218-227.

<Response> We thank the reviewer for their suggestion to enhance the Introduction and specifically to expand the summary of existing methods and comparison to the proposed method.  We have expanded the Introduction as suggested by adding a new paragraph that references and compares the two journal articles listed as well as several other articles to demonstrate the breadth of hyperspectral imaging technologies that are now being implemented in the medical imaging, and fluorescence microscopy fields.  We believe that the revised introduction is now more thorough and informative.

  1. Figure 1: please compare Fig. 1A and Fig. 1C in detail.

<Response> Thank-you for the valuable suggestion.  We have expanded the paragraph in the Introduction section to provide a detailed description of Figure 1 and to compare panels A and C, as well as panels B and D.

  1. Table 1: please list the imaging speed as it is important for fluorescence imaging.

<Response> We agree, this is important information for the reader.  We have added a line to Table 1 to list the overall spectral imaging speed.

  1. I think this paper lacks the discussion of spectral imaging requirements in context of fluorescence applications. For example, what are requirements for fluorescence imaging (spectral resolution, bandwidth, speed) and to perform successful unmix? What would enhance these techniques?

<Response> Thank-you for this suggestion.  A new introductory subsection has been added at the beginning of the Results and Discussion section that discusses spectral imaging requirements for fluorescence cell signaling studies.

  1. Section 2.3 Hyperspectral Image Data Pre-processing: please explain in detail how to normalize the wavelength data.

<Response> Thank-you for this suggestion.  This subsection has now been expanded to provide a more detailed explanation, with equation, of the hyperspectral image data pre-processing.

  1. Section 2.4 Reference Spectra and Spectral Library: please explain how to get the pixel-averaged spectrum.

<Response> Thank-you for this suggestion.  The details for extracting the pixel-averaged spectrum using ImageJ software have now been added.

  1. Figure 5: I think the results should be compared with the ground truth. I think the paper could be stronger if the performance comparisons have been clearly addressed.

<Response> Thank-you for this very valuable observation and suggestion.  Indeed, knowledge of a ground truth is important.  Unfortunately, for many biological cell and tissue samples, it is prohibitively difficult to generate an experimental sample in which the ground truth is absolutely known.  Because of this, we previously reported on the use of a theoretical sensitivity analysis framework to generate a hybrid experimental + theoretical HSI image dataset that contained known levels of a spectral signature mixed into an experimental image (see reference below).  This framework is useful for understanding the sensitivity and specificity of a spectral analysis algorithm for detection of a given target fluorescence signature in the midst of other confounding signatures.  In this study, we applied the theoretical sensitivity framework to estimate the accuracy of the four spectral analysis algorithms for identifying the Cal 520 fluorescent label, as shown in Figure 6 in the manuscript.  We then extended the theoretical sensitivity framework to allow estimation of a “minimum detectable limit” of the Cal 520 label, and used this minimum detectable limit as a threshold to define which pixels contained a meaningful level of Cal 520 vs. pixels that could be considered as background.  Pixels with meaningful / above the minimum detectable limit of Cal 520 signal were utilized for subsequent time-lapse analysis.  We have now added a new paragraph immediately before Figure 5 to better describe the limitations of creating a ground truth in live cell experiments and to outline the approach we have used in applying the theoretical sensitivity analysis framework.

Leavesley, S. J., Sweat, B., Abbott, C., Favreau, P. F. & Rich, T. C. A theoretical-experimental methodology for assessing the sensitivity of biomedical spectral imaging platforms, assays, and analysis methods. Journal of Biophotonics 11, 1–25 (2018).

  1. As the spectral analysis algorithms have been widely used, please explain what is new in the analysis method used in this paper.

<Response> Thank-you for the valid critique.  In this paper, several new aspects for these analysis algorithms, and the overall approach, are presented.  First, the analysis algorithms have never been applied to fluorescence excitation-scanning HSI microscopy data for measurement of dynamic cell signaling levels.  Hence, it was important to compare the performance of the analysis algorithms and to report on which may be best suited for this application.  Second, these analysis approaches are typically evaluated for fixed images and not time-dependent/dynamic image data sets.  Hence, it was important to evaluate the performance of these algorithms across the entire time series, which can be seen by viewing the supplemental video material. Third, the analysis algorithms were combined with our theoretical sensitivity analysis framework, which was then extended to develop a method to identify a threshold that would discriminate between signals above a minimum detectable limit and those below it, that could be considered as background.  This development is important as it allows dynamic cell signals to be recorded only from pixels above the minimum detectable limit, which should improve the reliability of the cell signaling measurements.

  1. A more comprehensive section of discussion is needed to address lots of issues, such as the advantages and disadvantages of the proposed method.

<Response> Thank-you for this very relevant comment.  Indeed, there are many advantages, disadvantages, and limitations that should be discussed when selecting an HSI microscope system and configuring the system parameters for a particular experiment.  We have added a new section, “3.4 Considerations for Live-Cell HSI Studies”, that provides a discussion of limitations and other considerations.  We hope that this additional information is valuable for readers as they consider how to adapt HSI approaches to their studies.

Reviewer 3 Report

I have had the pleasure of reviewing the manuscript titled "Comparing the performance of spectral image analysis approaches for detection of cellular signals in time-lapse hyperspectral imaging fluorescence excitation-scanning microscopy." This manuscript comprehensively investigates excitation-scanning hyperspectral imaging (HSI) for dynamic cell signaling studies, specifically the study of the second messenger Ca2+. The study demonstrates the potential of excitation scanning HSI with suitable spectral analysis algorithms and pixel filtering as a mechanism for measuring subtle kinetic events in live cell dynamic assays, which has significant implications for future research in this area. The time-lapse excitation scanning HSI data of Ca2+ signals are very impressive, and the authors have provided a thorough analysis using broad spectral analysis algorithms and adequate references. The manuscript reads very well and smoothly, and the introduction of theoretical sensitivity and cellular autofluorescence measurements is of great interest to researchers in the field. Based on my expertise and evaluation, I am pleased to accept the manuscript in its current format.

Author Response

We would like to thank the reviewer for their thoughtful review and very encouraging and detailed comment regarding the potential applicability of the time-lapse excitation scanning HSI data of Ca2+ signals.

Round 2

Reviewer 2 Report

I think the revised version is Okay to be accepted.